# Symbiosis or Sporting Tool? Competition and the Horse-Rider Relationship in Elite Equestrian Sports

**DOI:** 10.3390/ani11051352

**Published:** 2021-05-10

**Authors:** Rachel C. Hogg, Gene A. Hodgins

**Affiliations:** School of Psychology, Charles Sturt University, Wagga Wagga, NSW 2678, Australia; ghodgins@csu.edu.au

**Keywords:** equestrian sports, horse-rider relationship, equine welfare and management, elite sport, social constructionism, grounded theory, sporting performance, ethical equestrianism

## Abstract

**Simple Summary:**

Forming relationships with horses is commonly understood as an important part of amateur and elite equestrian sports. Horse-human relationships have historically been romanticized and the success of sporting horse-human combinations has been attributed to a close relationship between horse and rider. Elite equestrian sports differ from amateur equestrian sports, especially where an elite rider earns their primary income from riding and competing horses, and third-party sponsors and owners are financially invested in a horse-rider combination. In this study, 36 elite equestrian athletes were interviewed about their interspecies relationships. Results indicated that a strong horse-rider relationship could, in some instances, inhibit a rider’s ability to compete successfully and engage in a commercialized sporting context, due to tensions between an instrumental approach to animals and meaningful horse-rider interaction. Results also suggested that horse-rider relationships may be peripheral to performance outcomes, or conversely, essential to performance success. The relationship between sporting outcomes and the horse-rider relationship in an elite setting is clearly complex and multifaceted. An increase in transient, instrumental horse-rider relationships may be resulting in a shift towards a commercial, detached model of relating to horses, raising ethical questions around the professionalization of equestrianism and the management of competition horses.

**Abstract:**

The horse-rider relationship is fundamental to ethical equestrianism wherein equine health and welfare are prioritized as core dimensions of sporting success. Equestrianism represents a unique and important form of interspecies activity in which relationships are commonly idealized as central to sporting performance but have been largely unexplored in the sport psychology literature. Horse-rider relationships warrant particular consideration in the elite sporting context, given the tension between constructions of “partnership” between horse and rider, and the pragmatic pressures of elite sport on horse and rider and their relationship. The current study examined the link between sporting performance and the horse-rider relationship in an elite equestrian sporting context. Thirty-six international elite riders from eight countries and six equestrian disciplines participated in a single in-depth interview. A social constructionist, grounded theory methodology was used to analyze this data. The horse-rider relationship was positioned in three different ways in relation to elite sporting outcomes: as pivotal to success; non-essential to success; or as antithetical to success. Participants shifted between these positions, expressing nuanced, ambivalent attitudes that reflected their sporting discipline and their personal orientation to equestrianism. Competitive success was also defined in fluid terms, with participants differentiating between intrinsic and extrinsic markers of success. These findings suggest a complex and multifaceted connection between interspecies performance and relationships in elite sport. Where strong horse-rider relationships are antithetical to performance, a threat to the welfare and ethics of equestrian sport exists. Relevant sporting governing bodies must attend to this problem to ensure the centrality of animal welfare, wellbeing, and performance longevity to equestrian sports.

## 1. Introduction

Equestrian sports represent a particular challenge to sport psychologists, animal ethicists and behaviorists, as well as the riders and trainers who engage with horses in a competitive context. The complexities of equine management in sport are intrinsically linked to horse-human interaction, especially given the interconnections between human-horse relationships, interspecies communication, and attitudes towards animals. Sports such as dressage, three-day eventing, showjumping, and endurance sports are not publicly scrutinized ethically to the extent that flat- and jumps-racing equestrian sports are, yet risk management, alongside animal health and welfare, are critical components of ethical equestrianism. While issues of equine welfare and management are relevant to all levels of equestrianism, specific questions may be raised in the context of elite equestrianism, a context limited to horse-rider combinations who represent or have represented their country internationally and/or are listed on the elite squad for their country. 

Equestrian sports are unique and liminal within international sport, constituting the only multi-species sport currently included in Olympic level competition [1]. The marginality of equestrian sport within this framework has been highlighted in substantial recent changes to the format of Olympic-level equestrian competitions [2]. This, alongside a number of features of the elite sporting context, raises questions around horse welfare and the connection between the horse-rider relationship and sporting outcomes [3]. Given the centrality of the horse-rider relationship to all aspects of equestrian sport, it is critical to examine how this relationship functions within an elite context to understand the ways in which performance outcomes and animal welfare concerns intersect. This paper will examine the reciprocal influence of the horse-rider relationship on sporting performance for elite equestrian dyads, with constructions of performance and success problematized and examined in relation to the social relationship of the horse-rider dyad. As will be argued, the relationship between horse and rider and sporting outcomes in an elite setting is complex and multifaceted. Beyond objective measures of ‘success’ lie a plurality of definitions, and these definitions have implications for the ethics of equestrianism and the management of elite horses. 

### 1.1. Social Relationships in Sport

Effective equine management in a sporting context is not currently facilitated by the field of sport psychology, insofar as sport psychology has historically neglected relationships and focused on performance outcomes [4]. Both the American Psychological Association [5] and the Australian Psychological Society [6] describe “performance enhancement” as central to the work of a sport psychologist, while neither refer to relationships as critical to the athletic context. This notwithstanding, even non-dyadic sporting disciplines inevitably involve interpersonal relationships and social processes and are commonly built around organised social activities [7]. A number of researchers have called for greater recognition of interpersonal relationships in sport [4,8,9,10,11], and this may be particularly critical for interspecies dyads where the relationship under consideration is less clearly reciprocal and may be implicated in the way in which equestrian athletes behave towards and manage their equine counterparts. 

Social relationships are instrumental in enabling the achievement of goals, learning, and knowledge transmission, allowing us to express and develop a unique sense of self and identity [12]. Relationships that occur in a sporting context may also carry important implications for psychological functioning beyond the sporting context [13]. This may be particularly the case in the elite context where athletes’ lives are deeply embedded in their sporting practice. Social relationships are also relevant to understanding experiences of security, power, and unity [14], with implications for modern sporting practice. This point has been theorised in relation to equestrian dyads, with Shaw [15] describing, “the pinnacle of social coordination” as “unity…man on a horse becomes the horse-and-rider” (p. 158). As reflected within the field of social psychology, relationships are challenging to describe and understand, and interspecies relationships reflect a specific challenge for social psychology and the elite sporting nexus. 

### 1.2. Elite Equestrianism and the Horse-Rider Relationship

Research has addressed the general significance and meaning of the horse-rider relationship, and some studies have addressed the relationship between horse and rider in a competitive context, yet only a small body of literature exists concerning the relationship between horse and rider in an elite context. The elite sporting context contains many unique challenges for horse-rider dyads [16], and some of these dynamics raise questions around the meaning and significance, as well as the status, of the relationship between horse and rider in sport. While the transition to riding horses at an elite level may be gradual, working with horses at an amateur level is arguably different from what is expected of elite equestrian athletes. 

Several important dynamics of elite horse-rider interaction must be noted. First, elite riders commonly ride as professionals, meaning their primary source of income is generated from interaction with horses. As elite equestrianism becomes increasingly commercialised [16], this pressure is arguably increasing. An increased pressure to achieve performance outcomes quickly and in ways that may be antithetical to the formation of a strong relationship between horse and rider has arguably begun to pervade modern equestrian sport [16]. This has the potential to lead to compromises in horse training and welfare [17], while equestrian sports are under increasing scrutiny with respect to horse welfare and safety [18,19]. Not all equestrians choose to ride professionally, however, making it difficult to identify the exact number of elite riders who work and train horses on a full-time basis. Rudolph [20] notes that while some devote their lives solely to equestrian sport, others engage in paid work outside of sport. Given the costs associated with riding, keeping, and competing horses, it is not uncommon for equestrian athletes to attempt to make a career out of their sporting involvement. Indeed, the cost of keeping a single horse per year can equal or exceed $25,000, while elite horses may commonly cost six figures, with Coulter [21] estimating that elite showjumpers may cost anywhere between $150,000 to several million dollars. 

Second, while participating in equestrian sports at any level may mean jeopardising the welfare of one’s horse should accidents and injuries occur, elite horses are typically placed under greater mental and physical pressure than any other category of horses ridden for sport or pleasure [22]. The horse-rider relationship has been heralded as critical to performance success [23,24,25,26], yet the connection between these factors and horse welfare deserves careful consideration, particularly in an elite sporting context. Third, elite riders may engage with a large number of competition horses over their lifetime, and in some instances, these relationships may not be entirely within the rider’s control. For example, where a horse is worth a large sum of money and/or is owned by a third party or several third parties, these connections may play a significant role in determining the duration and nature of the relationship between horse and rider. Many elite horse-rider relationships have the potential to be highly transient, and this may change the nature of the relationship and/or the extent to which a rider feels equipped to invest emotionally in the relationship. Sponsored riders and those participating in high-level competitions, particularly team competitions, may feel a sense of obligation to their teammates and sponsors, which may in turn change the ways in which they engage with their horses. In his autobiography, renowned British eventing rider William Fox-Pitt [27] describes his experience of this shift, exemplified in his description of his unwell Olympic mount in the lead-up to the Beijing Olympics: “If this had been a normal competition…I would not have even put him on the lorry [to go to the competition]” (p. 286).

What takes place at the elite level is not only unique from the amateur context, but also inevitably influences what is considered acceptable at an amateur level and within the non-sporting arena. Elite equestrian sports are the most likely form of equestrianism to be televised and consumed by the general public, making horse-rider relationships in elite sport more visible than any other form of equestrian sport, while amateur riders aspiring towards elite status are likely influenced by the standard set at higher levels of the sport. 

The horse-rider relationship is widely considered fundamental to competitive successes with horses, yet a number of factors influence elite sporting success and media representations of equestrian sport do not always represent these complexities. In contrast, horse-rider relationships are often magnified and romanticized, while capitalist influences on the relationship are minimized. In an interview for the Fédération Equestre Internationale (FEI) [28], (35:00–37:00 min), Chef d’équipe for the American Reining Team, Jeff Petsaka states: 

“They [elite riders] invest a lot of time and money and emotion in these horses. You have to keep in mind that we all started this just because we love horses and that doesn’t change, it[‘s] just, the level at which we compete might change, but at the end of the day, the thing that binds us all together…is just the fact that when we were little kids we just couldn’t stay away from them [horses].”

These sentiments highlight consistencies in motivation for engaging in equestrianism, but overlook the role of contextual factors in shaping horse-rider interaction and how horses are managed across contexts, according to their financial and personal ‘worth’. Overlooking the influence of such structural factors, including but not limited to the professionalisation of elite sport, is to ignore an important aspect of equestrianism. It is commonly expected that riders engage with horses, ‘for the love of horses’, a position indicative of a personal relationship between horse and rider that develops naturally due to a passion for horses on the part of the rider. However, amid such romantic notions of horse-rider dyads, it is also widely acknowledged that elite riders commonly engage with a large number of horses both at single points in time and across their careers and may not have full control over the management of these horses. They may also have significant financial obligations to owners, sponsors, and the sporting organisation(s) under which they compete internationally. Navigating a career contingent upon continued competitive success may present important and unique challenges for these athletes and the animals with whom they work. 

### 1.3. The Status and Ethics of Elite Equestrianism

Elite equestrian sports occupy an intriguing position between business and recreation, crossing the boundaries between sport, hobby, leisure, and work [29]. Elite equestrianism is highly goal oriented [30], and while ‘sport’ crosses the boundaries between recreation and work, research suggests that professionalism and the ability to run an effective business are critical attributes for success in elite equestrian sport [21]. This emphasis on a business model of sport is important to note. Historically, sporting practices have been influenced by their militaristic origins as well as amateuristic sporting values [16], making for a nuanced sporting environment replete with contradictions. While lucrative prize money is offered in some circumstances, such rewards are offset against the cost of entry fees, horse stall boarding fees, and travel expenses, keeping in mind that only a small portion of riders will be awarded prize money [20]. The business of equestrianism is further influenced by the longevity of the human athlete, with many riders competing longer and later into their lives than athletes in sports such as swimming or gymnastics, for example [31], making equestrianism a possible source of lifelong employment for many elite riders. 

In another challenge to the horse-rider relationship, somewhat paradoxically, competitive success may serve to jeopardise the future of elite equestrian dyads, as successful horses are more likely to change hands, and riders, following a period of competitive success [17]. This was relevant in the case of renowned British dressage combination, Charlotte Dujardin and Valegro, with Valegro’s financial value skyrocketing to an estimated six million British pounds following their international success, creating enormous pressure on the owners to sell the horse lest his commercial value should unexpectedly decrease due to injury or a sudden decline in performance [32]. Although Dujardin previously expressed a pragmatic and accepting attitude with respect to this possibility, she has since revealed the tremendous stress she experienced facing the “impending loss” of Valegro [32] in the lead-up to the London Olympics. In an article with The Guardian, Dujardin (as cited by Kessel [32]) noted, “Every competition I do I never know if it’s the last one [with Valegro]…It’s always trying to put that to the back of your mind and carry on….” It is not uncommon for elite equestrian athletes to face such uncertainty. Mark Todd [33] describes a similarly stressful period following his gold-medal winning performance with Charisma, while much has been written of the separation of Dutch dressage champions, Edward Gal and Totilas.

Another concern for elite equestrian athletes is the demand for training and/or competition horses to be produced quickly, while supply-and-demand ratios for high-level performance horses are such that large sums of money may be made from a promising young horse, as well as a successful competition horse [17]. As Heuschmann [17] states, in business “time is money” (p. 18) and for equestrians, time also represents risk. It is in this context that Heuschmann [17] writes critically of riders who exploit horses due to an immense pressure to be successful, but is sympathetic regarding the financial and performance-based pressure elite athletes face, noting the dedication and sacrifice required, while also highlighting the difficulty of making a financially viable career out of equestrianism. 

Figures presented by the American Horse Council Federation (AHCF) [34] suggest that the equine industry involves over 7.2 million horses, has an employment impact of 1.7 million jobs, and contributes around $122 billion to the American economy. The European Horse Network [35] reports equally significant figures, estimating that horse-related endeavours contribute over $100 billion to the European economy on a yearly basis, while Australian data suggests that equestrian sport alone contributes over $1.143 billion to the Australian economy on a yearly basis, with 76% of Equestrian Australia members indicating their involvement in the sport centres around “the love of the horse”, with community members spending approximately $371 million a year on maintenance and transport of horses alone [36].

With increased financial investment in non-racing equestrian sports, the meaning and value of both the horse and sporting success may change in important ways for elite competitors. A failure to maintain one’s competitive profile has the potential to jeopardise financial deals. As an example of the commercial value associated with being a high-profile elite equestrian athlete, Rudolph [20] notes the sponsorship arrangements showjumper Rodrigo Pessoa, and dressage rider Anky van Grunsven have had with companies such as Rolex and Volkswagon. Although equestrian sports constitute an objectively highly expensive pursuit [37] and even elite-level riders may struggle to remain financially viable [27,33,38,39] high levels of success can result in lucrative sponsorship arrangements, particularly in the showjumping arena [40]. This may create a disjuncture not easy to resolve. As Rudolph [20] notes, “the tension between doing whatever it takes to win and ensuring a clean victory can stretch any competitor to the breaking point” (p. 47).

### 1.4. Performance and Relationships in Equestrian Sports

Research suggests that the horse-rider relationship strongly influences performance outcomes, with both Pretty and Bridgeman [23] and Visser et al. [25] arguing that the success of horse-rider dyads depends on the rider’s ability to establish a relationship based on mutual trust, communication, and submission. Yet as Visser et al. [25] contends, evaluating the interaction between horse and rider is a difficult task and therefore establishing a link between the relationship and performance is also complex. From a physiological, biomechanical perspective, a number of performance monitoring strategies have been operationalised to assess equine performance, though such strategies have largely been confined to horse racing, and are not necessarily applicable to all equestrian disciplines, particularly those where ‘performance’ does not involve strenuous physical exertion on the part of the horse. In a seminal study, Evans et al. [41] identified a relationship between the blood lactate response of a Thoroughbred racehorse to submaximal exercise and racing performance, finding that 48% of performance variability is reflected in the lactate response shortly after exercise. More recent research has found that infrared thermography may provide additional information relevant to monitoring equine performance in equestrian sports where maximal equine exertion is required [42]. Such physiological approaches to understanding performance in equestrian sports provide valuable information but typically focus exclusively on equine physiology, rather than considering horse-rider dynamics simultaneously, and may not be applicable to less strenuous equestrian sports such as dressage, para-dressage, and vaulting. In a scoping review of performance determinants in the sport of dressage Hobbs et al. [43] addressed the physiology and biomechanics of horse and rider and found that the gait of the horse, and posture of the rider, were central to performance. This suggests that minimal disruption on the part of the rider, alongside the intrinsic physical characteristics of the horse and training practices that reinforce correct gait and movement dynamics, are important to performance outcomes. 

It is critical to consider the subjectivities of performance in equestrian sports such as dressage and acknowledge the impact of such subjectivity on research findings addressing the relationship between performance and the horse-rider relationship. To some extent a strong, positive relationship might be expected between the horse-rider relationship and performance attainments. However, as noted by Stachurska et al. [44], there is often great variation in the opinions of different judges as to what constitutes a technically correct performance in equestrian sporting domains such as dressage. This has led to ambivalence around some sporting results, and questions as to whether performance results reflect a true harmony between horse and rider, while the extent to which equestrian judging aligns with the ethical prerogatives of the FEI has also been questioned [45,46].

Well established, compatible horse-rider relationships have been espoused as critical to competitive success in equestrian sport [47,48], yet certain equestrian sporting practices may run counter to these prerogatives. The history of elite equestrianism suggests that riding unfamiliar horses to international victory is a legitimate possibility, bringing into question the role of the relationship between horse and rider in equestrian sport. The practice of riding another competitor’s horse in competition, known colloquially as ‘catch-riding’ has historically pervaded a number of equestrian sports, while in McKernan’s [47] research on eventing riders, the average equestrian dyad had been working together for less than three years. Such practices are not unique to the elite stratosphere. Indeed, as a practice, catch-riding is not dissimilar to what occurs in the collegiate equestrian system in the United States, where horses are shared interchangeably among riders in an instrumental fashion, a kind of co-operative interaction between horse and rider focused on the achievement of specific, limited, and typically immediate goals.

These complexities notwithstanding, the relationship between horse and rider has been held as highly important to competitive success [17,33], as well as a source of motivation for involvement in equestrian sport [37], with a range of research [47,48,49] along with anecdotal and autobiographical reports [27,39,50,51] centralising the horse-rider relationship as a fundamental component of equestrian sporting participation. Former World Champion, Olympic medallist and six-time winner of Badminton Horse Trials, Lucinda Green [52] states of the horse-rider relationship in sport: “…confidence, harmony, partnership, call it what you like…it is difficult to pinpoint its ingredients, but it is easy to recognise when its presence is lacking” (p. 52). In research on professional riders, Smart [53] identifies communication, conflict, and powerful emotions as salient aspects of horse-human interaction, while Pretty [54] states that a mental connection with one’s horse is necessary to succeed in equestrian sports. This may be true, and Green [52] may well be correct when she suggests it is possible to identify when a relationship between horse and rider does not exist, yet the connection between horse and rider is nonetheless difficult to evaluate, particularly in the context of subjective sporting performances [44].

### 1.5. Equine Welfare in Elite Sport

From a welfare perspective, the elite competition context has been linked to activities of a maximum degree of strenuousness, acute/chronic horse injuries resulting from strenuous challenges, and long-distance travel to highly unfamiliar and therefore stressful environments [55]. Some aspects of equestrian sport build upon the horse’s ‘natural’ instincts, others do not; and may therefore represent a source of threat and stress to horses [56]. From a physiological perspective, research indicates that competitive riding may have a notable impact on the stress experiences of performance horses. As an example, Witkowska-Piłaszewicz et al. [57] found that blood cortisol concentration, a hormonal indicator of stress in horses, increased after both training and competition in a sample of performance horses, with the most intensive responses observed after competition. While an increased cortisol response is not necessarily indicative of welfare concerns, such findings suggest that competition events may evoke stronger stress responses in horses and therefore elite sporting horses are uniquely exposed to stress. This finding is corroborated by Art and Lekeux [58] in their analysis of endogenous and exogenous stress factors experienced by competition horses, while Bartolomé and Cockram [59] extend these findings to provide a summary of the key stressors horses experience in sports such as endurance, racing, eventing, showjumping, and dressage. Documented stressors affecting horses in these disciplines relate to a broad array of negative stress-related outcomes, including general equine health concerns as well as concerns relating to equine morphology, behavioural changes, and genetic factors.

Sport is an inevitably human-driven pursuit [60], though as Dashper [3] contends, this does not automatically mean that equestrian sport is exploitative, but is suggestive of the potential for exploitation, leading to questions about the boundaries between the use and abuse of horses in sport [18]. The ‘win-at-all-costs’ mentality, although not necessarily widespread within equestrian circles, has manifested itself in shocking cases of horse abuse [61] that serve to jeopardise the future of equestrian sports, the reputed ‘bond’ between horse and human, and most importantly, the welfare of horses involved in the human pursuit for sporting success [19]. Furthermore, competitive equestrian sports have been blighted by a range of practices with possible welfare ramifications for horses, including hyperflexion amongst dressage horses [62] and soring [63] amongst gaited horses, the practice of excoriating the skin using chemicals to encourage exaggerated movement [64]. Despite being illegal under federal law, practices such as soring continue to pervade some equestrian sports [61]. Another particularly grievous practice is that of using chemicals or sharp objects to heighten the sensitivity of the horse’s legs while showjumping [19]. In one extreme case, Sneed [61] notes a competitor was suspended from showing and judging Arabian horses after it was found that seven of his horses had undergone cosmetic surgery aimed at improving their appearance based on competitive ideals. 

The commercialisation and professionalisation of equestrian sports presents a challenge to horse welfare and the horse-rider relationship [65]. As Bridgeman [49] notes, successful elite horses are now a multi-million dollar commodity item to owners, breeders, and other investors in equestrian sport. Elite horses may be cared for using state of the art technologies and feeding products, yet Henderson [66] argues that basic psychological wellbeing may be compromised by the very practices designed to ensure an elite horse’s safety, wellbeing, and continued performance success. The risk of injury to horses worth large sums of money means that many elite competitors choose to house their horses in intensively managed, restrictive environments [65] and although safer than more natural arrangements, such living conditions may have deleterious welfare implications for horses [66] as well as spin-off effects for the horse-human relationship [67]. In addition, as Endenburg [30] notes, demands are being placed on competition horses at an earlier age than ever before, meaning that the horse may not be physically fully developed by the time its competitive life begins. Derksen and Clayton [68] argue that many competition horses never reach their full potential due to inappropriate training or management practices that result in injury or horse-rider conflict. 

An assumption inherent in many sporting domains is that the safety of horses lies in the hands of a “knowledgeable and caring rider”(p. 1259) [69], and less so with veterinarians or other parties who have fewer opportunities to assess the horse’s state. Public opinion plays an important role in determining how elite sport proceeds [18] and riders are key stakeholders in upholding and prioritizing equine welfare [17], yet questions may be asked regarding the level of agency riders have in ensuring the welfare of their horses is upheld. A number of elite competitors have experienced a fall during competition that resulted in the death of their horse [27,33,39,70]. As an example, well-known New Zealand eventing rider, Mark Todd [33], describes his anguish and self-blame at the death of one of his horses during a cross-country round, with similar emotions expressed by other riders in the same circumstances [27,39]. Todd also suggests a broad lens of responsibility is required, arguing that organisers and course officials must be held responsible for the conditions under which a horse is placed in competition, stating, “No rider who is in a position to win a medal, or even to complete for their team, is going to canter around an Olympic Games” (p. 83). 

The sustainability and ethical foundations of elite equestrian sports depends in no small part upon the value attributed to the horse, and the relationship between horse and rider. It is imperative that this relationship be examined and the connection between this relationship and sporting outcomes carefully scrutinised. While this is important at all levels of competitive equestrian sport, elite athletes exist in a professional, economic context that places a greater degree of pressure on the horse-rider relationship. Elite riders also represent the public face of equestrian sports and may profoundly influence other equestrian cohorts, hence the need to understand the experiences of this cohort. 

The aim of the current study was to explore the relationship between horse and rider in an elite sporting context and to construct an analytic framework of this relationship. Two key research questions were used to guide this process of inquiry. First, what characterises and defines the relationship between horse and rider in an elite sporting context? Second, what are the implications of the horse-rider relationship for competitive performance in elite equestrian sports? Critical to this inquiry was a need to understand how ‘competitive success’ may be defined and understood in an interspecies sporting context. These research questions created a theoretical scaffolding through which to explore the ways in which horse-rider relationships are developed, framed, challenged, and understood by human athletes immersed in the sporting context in which such relationships take form.

## 2. Materials and Methods

### 2.1. Study Context

This research was a part of a larger qualitative project aimed at generating a substantive theory of horse-rider relationships within elite equestrian sport. This explication of that theory focuses upon the impact of (elite) competition on the horse-rider relationship. It is also important to contextualise this study within the academic framework from which it was produced, namely, that of psychology. Extant findings, theories, and ideas, particularly from sport and social psychology, were acknowledged and problematized and influenced the development of the research. These concepts did not always fit neatly with the phenomena under consideration, however, in part because of psychology’s ambivalent relationship with animal lives and scant literature around interspecies sport. Psychology has long neglected human-animal interaction. It has also privileged the mind over the body, in keeping with Descartan tradition [71], making physically oriented social relationships an intriguing, often devalued area of inquiry [53]. As such, the study of equestrian sports represents a challenge to those researching relationships in sport and to many of the assumptions implicit to psychology as a human-centric, individualist domain of inquiry. Engaging with and resisting these narratives were important in developing new theoretical insights around horse-rider interaction. 

### 2.2. Theoretical Approach

Symbolic interactionism was applied and extended in the current study as a lens through which to understand horse-rider engagement. This theoretical paradigm offered a way of approaching the phenomena in question and drew into focus a particular set of concerns regarding subjective meaning and action in relationships. According to symbolic interactionism, society, reality, language, and ‘the self’ lie at the heart of social interaction [72], with meaning constructed socially through dynamic acts of interpretation [73]. Not all aspects of symbolic interactionism centre social processes, however, individuality and agency are paramount in symbolic interactionism. Human beings are positioned as capable of defining, using, and changing their environment according to their understandings of it [74] and herein tensions emerge between symbolic interactionism and the study of human-animal interaction. Symbolic interactionism is ‘human-centric’ in every sense. Selfhood, language, and the mind are privileged; seen as transcending all other forces and influences [75]. The mind contains intelligence, knowledge, and culture while the body is ‘functional’, but not meaningful [76]. 

Engaging with symbolic interactionism required a reformulation expressly directed towards resisting the anthropocentric orthodoxy [77] of this theoretical perspective. Such a reformulation represents a development rather than a distortion of the picture of social reality that symbolic interactionism can capture. As Schlosser [78] notes, intimate physical exchanges, such as those between horse and rider, contribute to the development of social relationships in salient, often difficult-to-articulate ways. Understanding the social processes that extend beyond species boundaries is important to psychological research inquiry [79]. The principles of symbolic interactionism may be incorporated into research on animals if researchers are prepared to consider mechanisms other than language through which a concept of self may emerge [80]. The current study sought to achieve this, cognizant of the inevitable tensions between symbolic interactionism and the study of non-human relationships.

Symbolic interactionism provided a means of amplifying and contextualising the world as it appeared in participants’ data [81]. Questions about what experience means to an individual, and how and under what conditions the meaning(s) of experience change, were provoked by symbolic interactionism. The ways in which participants described and positioned horses revealed important insights about their relationships with them. I was interested in the meaning(s) participants attached to competitive, elite-level horses and the ways in which this meaning was transmuted through horse-human interaction. Under what conditions does the horse-rider relationship become ‘problematic’ and if or when this happens, why does it happen? Elite riders typically interact with horses on a routine and regular basis, and it is likely that, as Charmaz [72] states, collective action leads to a lessened consciousness of action and the meanings attached to it [82]. Seemingly ‘universal’ discourses pervade competitive equestrian sport, facilitating collective action and thought, and these discourses warrant consideration.

### 2.3. Methodological Approach

A social constructionist qualitative research methodology was used to explore the intersections between performance and horse-rider relationships in elite equestrian sport. Qualitative research facilitates the exploration of social action and meaning via the collection of in-depth, rich data, contextualised within the social context from which it was drawn [83]. As such, this research approach aligned with symbolic interactionism and the focus of the current study. Social constructionist grounded theory is an inductive, iterative, open-ended methodology and method in which the researcher’s analyses are represented as constructions of the studied world [72]. Grounded theory seeks to examine the ways in which research participants “use language and form and enact meanings” (p. 95) through language [72] and this approach allowed the data of this study to be explored with attention to the subjectivity that Charmaz (p. 14) states “is inseparable from social existence”, including that embodied within interspecies social relationships. Social constructionist grounded theory requires a conscious examination of the preconceptions, values, and other factors that may limit, define, or shape what, as a researcher, I was able to identify in the data, drawing home the notion that research is constructed between researcher and participant, not simply emergent from participants’ narratives [72]. As Charmaz [72] notes, preconceived ideas and pre-existing knowledge are unavoidable qualities researchers bring to their work and should be considered as part of the conceptualisation process. 

### 2.4. Data Collection and Analysis 

A series of thirty-six in-depth qualitative research interviews were undertaken with twenty-nine elite riders, four former elite riders, and four sub-elite riders. As Charmaz [72] suggests, a major advantage of scientific interviewing is the opportunity for participants to express thoughts and feelings disallowed in other contexts and social relationships. The interviews also differed from positivist strategies for studying psychological phenomena, inasmuch as I was neither in control of nor distanced from my participants [84]. Participant recruitment took place through word of mouth and third-party contacts, as well as by contacting riders either via an email address provided on their website or by private messages on social media sites such as Facebook. The size of the participant sample supported the theoretical claims of the study, with a range, depth, and breadth of accounts providing a sound foundation for the analysis that followed [85]. Institutional ethics approval for this study was provided by the School of Psychology Ethics Committee of Charles Sturt University (ID: 2010/13). Pseudonyms were assigned to all participants.

Data analysis took place via an iterative combination of coding and comparing data, as well as developing codes and categories through an appraisal of sources beyond the interview data, in line with social constructionist grounded theory practices [72]. Connecting early codes provided an analytic skeleton through which I was able to begin considering what Charmaz [72] (p. 113) calls “nascent” theory, with categories of analysis developed through an analysis of micro-level codes. This nascent theory directed me to further questions about the data, analysed progressively throughout the study and with the use of texts such as autobiographies and memoirs from former elite riders. I attempted to invoke what Charmaz [72] (p. 116) refers to as “a language of action”, something which Clarke [86] suggests allows for the articulation of a process, but in abstract, sophisticated ways. Examining explicitly articulated interview data, as well as what was left unsaid by participants, and what may have been implied by these omissions, was central to the analysis process. Variation within early categories was identified by comparing data with data, and codes with codes. As an example, when participants were asked to describe a past performance that stood out to them, some described unsuccessful but personally satisfying experiences, while others described successful performances as their most meaningful experience, with subtle moral and identity claims implied in some but not all narratives. By attending to such variations in the data, the significance of performance outcomes for the horse-rider relationship became clearer over time and as the analysis continued.

### 2.5. Participant Demographics

Participants were selected based on their experience of having ridden competitively in one of the following FEI equestrian disciplines: dressage, eventing, showjumping, para-equestrian dressage, endurance, vaulting, reining, and carriage-driving. Participants were predominantly drawn from the three Olympic sporting disciplines of dressage, eventing, and showjumping, though a number of para-dressage, endurance, and vaulting riders were also interviewed. An attempt was made to recruit participants from a variety of equestrian disciplines in order to elicit a broadly meaningful, but also sufficiently focused participant sample, in keeping with King and Horrocks’s [87] position on diversity in qualitative interviewing. Table 1 provides some information on the demographic characteristics of the thirty-six participants who were included in the final data analysis.

### 2.6. Reflexive and Ethical Insights 

Reflecting on one’s position as a researcher may evoke otherwise unacknowledged realities. The first author’s social identity, values, personal history, academic background, ontological orientation, and past experiences of engaging with horses all shaped this research [88]. Though participants did not always enquire, many assumed or learnt over the course of participating in the study that a personal connection to the research existed for me, as researcher. It would be feasible to overestimate my own insight into the lives and experiences of elite riders because I too, have ridden horses. I have never competed at an elite level in equestrian sport, however, nor am I a professional rider. Furthermore, much could be lost and gained by participants identifying me as an ‘insider’ to equestrian culture and sports [89]. On one level, I am in certain respects an insider to equestrian sport, and this helped in establishing rapport and understanding participants’ experiences. Insider status is accorded to a researcher when they share experiences, a common identity, and language with the group under investigation [90]. It has been argued that commonalities lead to a sense of legitimacy in the researcher, can mean better rapport with participants, and a fuller knowledge and understanding of the research area [89,91]. Yet knowing more can also mean seeing and asking less of the data. I attempted to engage in continual reflexive awareness of self, not as an insider or an outsider, but from an intermediary position of shared experience and difference [89]. Whilst being familiar with equestrian sports gave me common ground on which to establish rapport with participants, I was also cognizant of the assumptions underlying much of the ‘in-group’ language in equestrian circles and aware that a certain distance between myself and participants might facilitate a deeper exploration of language and assumed meanings. I did not set out to emphasize the commonalities I shared with my participants nor did I try to distance myself from them. Instead, I attempted to distance myself from what I knew, or thought I knew, about the topic of the research. 

Certain ethical issues emerged throughout the study and were analysed reflexively. As an example, while collecting data, I became increasingly aware of the political dynamics underlying what riders disclosed about their relationships with horses. One participant confessed that he had not put much time into one of his horses prior to a recent competition, which he felt led to a poor performance and he knew disclosing this to the horse’s owners could have negative consequences for him as a professional rider. Respecting participants’ privacy was more complex than simply anonymizing the data; though particular attention was paid to these issues given the public profiles of the riders involved in the study. Consent is also an ongoing process in qualitative interviewing and was a critical part of the current study. Ownership of and power and control over the stories that participants shared was also important to reflect upon. Elite riders often speak not just for themselves, but for their brand, and in some cases, the integrity of that brand is monitored and controlled by their management and/or sponsors. The ways in which participants’ interview data were presented and used in this study also has potential implications for the image of equestrian sport, particularly in light of increasing ethical concerns around the sport. In asking participants to speak about their relationships with horses, I was asking them to reveal information that directly reflected on them personally and professionally. Participants were accorded a certain degree of control over the course, pace, and content of the interview sessions, but researchers have substantial control over what happens next. A reflexive research approach aided me in recognising, problematising, and attempting to resolve these questions of ethics and research practice as the research evolved.

## 3. Results

Participants defined the connection between sporting performance and the horse-rider relationship in three distinct but non-static ways: as pivotal to success, non-essential to success, or as antithetical to success. The sporting context and/or discipline of the participant, as well as the participant’s personal orientation to equestrian sport, influenced their perspectives. Participants shifted between these conceptions, expressing nuanced, sometimes ambivalent attitudes regarding the connection between performance success and relationships in sport. Success was also described in fluid terms, with participants differentiating between intrinsic and extrinsic markers of competitive success. These constructions and issues will be explored in the category “a blessing and a burden: bonding and being competitive”, while more micro-level analysis is presented in the code “investing emotionally: managing emotional (de)attachment to horses” regarding how emotions and (de)attachment impact upon competitiveness. Participants used the terms ‘relationship’ and ‘partnership’ interchangeably when describing the bond between horse and rider, and thus the following analyses draws upon both terms. 

### 3.1. A Blessing and a Burden: Bonding and Being Competitive

Many but not all participants described a strong horse-rider relationship as critical to their sporting success. Elite vaulter, Jessica, saw the partnership as synonymous with participating in sport, noting “if you don’t have a partnership with your horse then it’s not really an equestrian sport, the horse is just an object that you’re on”. Three-day eventing rider Georgina echoed similar sentiments, noting, “I don’t know how you do it [the sport] without [a partnership]”, while dressage and three-day eventing rider Jim described the partnership as synonymous with being successful and fundamental to authentic sporting participation:

“The thing that stops the sport from becoming commercial though, is that you can’t fake that, like it has to be a partnership in that you have to throw your whole soul into it, you can’t say, “Oh trust me trust me” [to the horse] really to yourself think “No way”. If you say to a horse “Trust me” it’s got to be that you’ve thrown yourself into the…zone without reservation…so it takes part of your life. And so producing horses at the top, it could be a lucrative business, but you only have so much life, you can’t produce it like a sausage factory…that stops the whole industry from becoming commercial because every horse produced out there are little bits of people’s souls.”

The physical challenges of sport were used as a way of conceptualizing the centrality of the horse-rider relationship to sporting success, illustrated here by former eventer and current showjumping rider, Lucy:

“We’re going to go out and jump that World Championship cross-country course out there, you want a pretty jolly good partnership with it and the same with those big jumps [rider’s name] is going to jump; you want a pretty jolly good partnership with your horse.” 

While Hannah struggled to see how she could be competitive against other endurance riders for whom horses carried little value, she nonetheless endorsed the partnership as important, but for reasons beyond competitive success. Rather, she argued that the relationship was important because of the large periods of time spent together in relative isolation on course during which “sometimes you carry the horse and sometimes he carries you”. Unlike fellow endurance rider, Lara, Hannah saw catch-riding as deleterious to performance:

“It’s also very difficult to umm compete overseas with us because it’s very difficult to take our horses abroad and then we have to hire horses and it’s not the same, you don’t know the horse and he doesn’t know you, and it’s just…you have to understand each other.”

The extent to which the relationship between horse and rider was seen as influencing sporting outcomes depended on how the partnership between horse and rider was conceived and understood. For Hannah, an ideal partnership was reciprocal and fair, yet she differentiated between affection-oriented partnerships and more mechanical forms of partnership in endurance riding, suggesting that “working together,” was necessary for endurance riding, but this did not mean a horse-rider interaction was “nice” or even necessarily ethical. Other participants defined the partnership in more straightforward ways, linking it to a sense of teamwork and mutual focus between horse and rider that naturally facilitated sporting success and enhanced performances, as para-equestrian athlete, Grace, illustrates: 

“It only works if the horse is working with you; if they start to work against you or if, if they are not trusting you I don’t think you’re ever going to look as bright and shiny when you compete.”

As well as contributing to performance success, a strong horse-rider relationship may be fundamental to avoiding sporting failures of the most lethal nature. A failure to engender the horse’s goodwill could have serious consequences for not just performance outcomes but horse-rider safety, as Grace suggests:

“…it’s [the horse] the stronger animal, that’s for sure. It will win once it’s against you so you so you better, don’t have six hundred kilos against you, you better work with them, you know.”

Some participants had successfully competed on horses with whom they were not familiar, but most described knowledge of the horse as a kind of competitive currency. The more time spent together, the better. A range of ‘successful’ relationships were deemed possible, including those of competition-oriented riders who viewed horses as objects. However, to achieve longevity, reach the highest echelons of sporting performance, and “bring up multiple horses over long periods of time” (Tegan, para-equestrian), a partnership was non-negotiable. Performances could occur in the absence of a partnership, but as dressage rider Elizabeth noted, to “excel…become more brilliant” required a partnership. 

Mechanical skill in some instances was positioned as more relevant than relationships at the upper echelons of the sport, but a distinction could be drawn between being “elite” and being at the very highest level of the elite sportsmanship category. For example, dressage rider Kelly endorsed the potential for elite performances to occur regardless of the status of the horse-rider relationship, noting, “I’m sure if you asked all the Grand Prix riders to swap horses I’m sure they would all be capable of swapping horses and riding a Grand Prix test”, but believed that “effective, top-level [horse-rider] combination[s]” possessed strong relationships. In contrast to this, other participants saw basic performances as impossible in the absence of a partnership. Interestingly, para-equestrian rider, Grace, drew upon her limited physical strength as a competitive advantage because it meant performances could only be achieved through partnership:

“I think what most of our riders do, talking about para-equestrian[ism], is that we do always see them [the horse] as a partner and we can never ever work against them, because we cannot use any force. If I use force I am sitting beside the saddle usually so that might be, at some point people might think that is a disadvantage but sometimes I think when I look at some other riders it might be an advantage.”

Interestingly, female participants articulated a similar set of arguments regarding physical strength, the partnership, and performance outcomes, with eventing rider Claire suggesting male riders could substitute force for goodwill, circumnavigating the partnership. A kind of exceptionalism also characterised some notions of partnership, with certain participants describing the partnership as a priority to them, but not to other riders, highlighting the range of philosophies that may drive elite riders. Georgina and Hannah were curious about the views of these ‘other’ riders, with Georgina alluding to one very well-known elite rider “who’s got so many horses I don’t know how he knows what he’s sitting on”. Good riders were those able to establish strong relationships with horses, while gifted riders were those able to perform on their physical ability or simply establish immediate connections with horses. Of the latter category, Lucy states, “It’s only exceptional riders that can get on horses and get a partnership and that’s because they’re not too hard on the horse and they just work with them, feel their way round them”. 

Rather than suggesting that “exceptional riders” dominate horses into submission, Lucy argues that a tentative approach coupled with high levels of ability may facilitate the establishment of an immediate partnership, changing the ways in which the rider may manage or relate to the horse. Georgina saw the partnership as “huge” in relation to sporting performance, but was cognizant of the repeat catch-ride successes of other international riders, noting:

“A relationship with those genius riders is probably less necessary, or it forms instantly…because everything works so perfectly for those good riders you tend to think they’re all automatons underneath but I don’t think they are, they happen to be beautifully trained, which makes them look as if they’re not existing on a relationship. Whereas those of us who are less skilled riders are more needing that relationship, that’s how I look at it.”

Although Georgina considered herself less skilled than some other elite riders, her exceptional riding history suggested otherwise, and her position here reflects the different ways successful elite riders may navigate their chosen sport. The partnership could also influence a rider’s ability to sustain a successful long-term sporting career. Given that large amounts of time and money are invested in equestrian sports, particularly at the elite level, a strong horse-rider relationship may provide a measure of security in an otherwise uncertain and psychologically challenging competitive climate. 

Knowing one’s horse well may also facilitate a competitive focus. As an example of this, showjumper Beth described experiencing a harmonious and connected relationship with her horse that allowed her to feel in control, confident, and clear about what lay ahead of her during her Olympic medal winning performance:

“I was never nervous like, I knew from the moment that I stepped onto my horse that we were gonna jump a clean round…there was no focus on placing or results …I only allowed myself to think in terms of…what things I could control…that’s the biggest mistake I see, people thinking “Oh this point, or if I make a mistake it’s this many points or if I jump clean I have this many points and I’ll be in this place or that place” and that is not part of, that has nothing to do with the partnership or the performance.”

The bond between Beth and Botany, as well as Beth’s knowledge of the horse allowed her to include his actions, alongside her own, within the framework of things she could control, facilitating a positive performance outcome. As Kathy suggested, in partnership one has a better “understanding of what makes them [the horse] tick” and this was an advantage in the competition arena. Indeed, Thompson and Nesci [92] found that equestrian athletes commonly manage the unpredictability and dangerousness of equestrian sport by developing relationships with horses in which familiarity with the horse serves as a form of risk protection. Given the many variables that affect the sporting outcomes of equestrian dyads, choosing to focus on the relationship between horse and rider could also serve a protective function when performances did not go well, and this helped to build a resilient approach to sport. 

Judging one’s performances according to personal, subjective prerogatives may be one way of guarding against the disappointments and uncertainty of sporting outcomes. Although she had been a highly successful, competitive elite rider, eventing rider Natalie’s ethos toward equestrian sport centred on training rather than performance success. This helped her to prioritise relationships over success, and horse welfare over financial gain, though she suggested that such priorities inadvertently led to success in many instances:

“For me it was always about the training, the competition is just the proof of the training…And if it happens that you happen to win, well and good, but you’re not out there to beat this one and that one, you’re not out there to win at all costs…with horses, there’s always going to be ups and downs and there’s always going to be disappointments and there’s always going to be a horse going lame and whatever but if your mentality is, “It’s all about the partnership, it’s all about the training” …you can go out and compete and not have any other competitors, it doesn’t matter, what matters is umm trying to do your personal best.” 

In some cases, a partnership facilitated success because as well as encouraging willing, unified performances, it enabled dyads to compensate for a lack of ability or talent on the part of the horse or the rider, as Tegan indicated with respect to her horse Sage: 

“The competitor in me would like…certain things in her physically…but I cannot ask for more as far as our partnership and relationship, I mean we have, that is nailed and umm it’s interesting because that is probably the thing that has made us so competitive, where I can go in the ring against very big, fancy flashy expensive horses…. but because we have that sort of harmony thing going for us…we look like one unified piece and we’re very accurate in the way we compete and that’s how I can beat those big horses and umm and so I would say you know that if I was just riding her and we didn’t have that, I probably wouldn’t be doing as well on a horse like her.” 

In the same vein, para-equestrian Ellen saw the bond between horse and rider as critical to performance success, noting how both her Grand Prix horses “far exceeded their...talent, because the relationship was there”, while Georgina described the success of one of her eventing horses as the product of his “heart” rather than his “ability” describing how she fell “deeply in love with that sort of heart”. The horse-rider relationship could also make a better rider, as para-equestrian Val described:

“I wouldn’t be able to ride as well as I can now if I didn’t have Moss. He has made me what I am, ‘cos I’ve ridden lots and lots of different horses but nowhere near, made nowhere near the progress on all those horses that I’ve made in the short term with Moss.”

The sum of horse and rider together may equal more than the sum of either individual. Referring to several well-known elite riders who shared close relationships with their current mounts, Tegan described how certain horse-rider combinations possessed a connection that made them highly competitive, suggesting that horse-rider compatibility may in fact play a role in the partnership and in sporting success. Dressage rider Abbey also suggested that a horse may be brilliant, but with a different rider the horse may not be so successful and vice versa. Building a partnership may be directly related to performance outcomes and the fulfilment of one’s athletic potential.

A direct link between sporting success and the horse-rider relationship did not always exist. However, with a number of factors influencing sporting success participants were careful to contextualise the horse-rider relationship as just one of a number of ingredients essential to achieving success in equestrian sport. A partnership could influence performance outcomes, but it did not guarantee success or eliminate other factors that also impact performance, with other dynamics sometimes as or more predictive of success than a strong relationship. When all other factors are held constant, the relationship between horse and rider may be pivotal, but as a number of participants noted, a wide range of factors influence performance outcomes, including the talent and physical aptitude of horse and rider, environmental factors such as course conditions, sporting politics, and discrepancies in judging. In context to this, Grace noted, “I don’t know if judges always see it [the partnership], that might depend on the judge…”. Environmental and social factors outside of the horse-rider relationship could also influence performances, with vaulting rider Hayley noting how past injuries, difficult weather conditions, and a troubled relationship between herself and her lunger had contributed to several recent less than ideal performances. 

Vaulter Claudia believed that a strong partnership could improve performance, but saw strong sporting performances as possible without a partnership, while para-equestrian Kate noted, “It’s never gonna work no matter how good the relationship is…if they’re a big striding horse and especially in para [-equestrian], if the rider, para rider can’t sit to that kind of movement”, though other participants suggested a connection between horse and rider could compensate for physical incompatibility. Although Diana believed relationships with horses were central to sporting success, she was quick to highlight “other things” that affected performance outcomes in elite dressage, particularly “politics in dressage”. Participants also emphasized the challenges of equestrian sport, even where a strong horse-rider relationship existed, suggesting that a partnership between horse and rider and talent did not guarantee sporting success. Elizabeth experienced a strong relationship with her current competition horse and together they had achieved high levels of success, but this did not prevent them from making two mistakes during a dressage test just prior to our interview. Other participants separated the relationship between horse and rider from the true “work” of equestrian sport, as illustrated here by Claudia:

“I think if you have a really good partnership with your horse, it can like, help your performance in the competition but I also think, even if you don’t, you can still have a really good performance if you don’t have the partnership but if the horse does its job and you do your job.”

Interestingly, Lara saw a partnership as critical for other sports, but was ambivalent about its influence on her own sport of endurance:

“A lot of times in the sport like I said I just get on a horse and go and try and build a rel[ationship]…those can’t be built in one day, they can’t, so it’s more like building an understanding. But you do see success in the way of…[rider’s name] on her horse [name]…she’s been riding that horse for a while so she has success in the way of, she knows her horse so well, she knows how fast she needs to go…I’ve seen people do really well on a horse they’ve never ridden before and that’s probably the really good work of the owner, whoever conditions the horses.”

Resisting the pull to establish a partnership quickly, Lara saw herself and other endurance riders as able to operate competitively in the absence of a true partnership, emphasizing the physical training over an interpersonal connection between horse and rider. Yet potential existed for the partnership to heighten or enhance competitiveness. Similarly, Claudia believed vaulting performances could occur in the absence of a strong relationship, but she was wistful about the pleasure of owning a horse and being able to spend time with it, noting how getting to know a horse enhanced her capacity to predict the animal’s behaviour. Ethan too felt that in some instances, talented eventing horses will perform well regardless of who is riding them, describing these horses as “triers,” however, he believed that less talented horses may also be successful by virtue of “incredible partnerships with their riders,” that “works because they know them so well and they’re advantaged by having such a good partnership/relationship with that horse”. A strong horse-rider relationship may also be seen as a buffer when faced with the pressures of performing to an audience, as Jane describes:

“If you’ve got that confidence and something does go wrong in the test, the wheels don’t fall off, whereas if you’re not 100% umm in that relationship with each other and something goes wrong then immediately you’re at each other.”

Participants also believed that in some instances sporting performances could be elicited through force rather than partnership. Natalie states, “It can be [a partnership] or it can be a dominating... with a lot of boys like well, fifteen-year-old girls too, it’s a domination thing”. Natalie also emphasized the importance of achieving performances through a horse’s willingness, suggesting that performances may be elicited “dishonestly”: 

“I often say to kids, “The horse has to do it because it wants to do it for you, not because it’s scared of what’s going to happen if it doesn’t do it.” That’s always been my way, I mean, that’s not everybody’s way, and certainly when you short-cut and you want results umm before the basics are really...before the basics are really instilled in the horse then the only way you can get them over the fence is just bashing them around and to me, if you have to bash your horse around a course, it shouldn’t be there, you know, some horses like to do it, and they want to do it for you, other horses really don’t have the talent for it, don’t have the ability for it, or are frightened, and if you bash them over a fence, it’s not fair.” 

Natalie raises a critical ethical point about equestrian sport, namely the potential to abuse a horse’s willingness. Horses may be large, powerful animals, yet they are also vulnerable to domination, and their sensitivity to fear as a preyed upon species provides an avenue for compromising the integrity of horse-human relationships by eliciting performances in the absence of trust, confidence, respect, and harmony. As Natalie suggested, partnerships were not the only way to achieve sporting goals or the only way to interact with horses, even if they may be the ideal means through which sporting goals are achieved. While Melissa believed the partnership was critical to sporting performance, she had observed other ways of operating as an elite rider and managing horses:

“I know there’s certainly a lot of people out there... say, [rider’s name], successful, and they run a million horses…the place is crazy. He’s very successful but he goes through a lotta horses, and he goes through them like a factory and umm I don’t want to be like that.”

Producing large numbers of horses through mercenary means may work in the short-term but was not typically conceived as sustainable across a lengthy career with horses. As Elisa stated, “Some people think you can force a horse, and maybe you can for a short while, but not for a long period”. While Georgina regarded her own performance successes as a product of the partnerships she was able to build with horses, she was equivocal about the influence of the horse-rider relationship on sporting success generally, noting how “domination works to a degree”, though she believed domination was a less broadly functional approach to achieving success. Beth expressed similar sentiments: 

“I do think a strong willed rider can, can dominate and and exude a performance out of an animal that maybe doesn’t have the best partnership yet or is developing or is not a partnership…a strong-willed rider can get that out.”

Participants referred to “other” riders successfully catch-riding or “exuding” performances out of horses, but for them, a partnership was essential to success. Yet a number of participants were competing at the highest level of their sport on horses with whom they were not yet well acquainted or with whom they did not share a strong relationship, while Ellen even described her most memorable sporting performance as winning a freestyle dressage competition with a “draw mare” she had ridden only ten times prior to competition, noting “we just clicked …it was spectacular”. While Ellen’s experience appeared an exception to the rule, other participants, such as dressage rider Crystal, were working successfully with horses with whom they had only recently become acquainted, and some were currently competing horses with whom they were not working well. Emma described her relationship with her current competition horse as “miles apart” from what she considered an ideal partnership, while Ellen expressed similar sentiments about her current international horse. Such less-than-ideal partnerships may have curtailed the performance potential of the dyads in question, but it had not stopped them from competing at the highest levels of equestrian sport, a finding suggestive of the complexity of the relationship between dyadic harmony and performance success. 

The capacity to form relationships with horses quickly, despite the emphasis participants placed on building relationships over time, may serve an important protective function for elite equestrian athletes, for whom being consistently successful is typically critical to ensuring a livelihood. When Grace’s horse unexpectedly injured a ligament, she was able to pick up a different horse and win a European selection trial for a major international competition only five weeks after first riding the horse, with the pair continuing on to compete internationally in their field, though as Grace noted, their relationship was still developing. The partnership between horse and rider was commonly understood as an evolving entity, leading to a non-linear connection between the partnership and sporting success. Georgina, for instance, described a difficult relationship with Morris, a “very very tricky” horse with whom she had experienced the highest levels of success. Despite their success, the interpersonal difficulties between the pair did not abate over time. Notwithstanding the emphasis Georgina placed on trust and partnership, she still described a challenging relationship with a horse that “used to hit everything, I nearly fell off, he just kept getting up the grades, but he was a liability I thought”. She noted, “I just never really knew what made that horse tick”, yet this did not prevent the pair from competing and winning at the highest level of their sport. 

While a partnership was often espoused as essential to sporting success, or at least one of a number of factors that may influence performance outcomes, participants also engaged with the notion that a personal attachment between horse and rider could be an impediment to success. The most compelling example of this came from endurance rider, Hannah, whose main competition horse meant a great deal to her. This for Hannah was a competitive disadvantage when competing against riders who did not value the lives of their horse(s). A strong relationship meant that “competitively…you understand your horse better”, but this did not necessarily offset the benefits of a more mercenary approach to horses in endurance racing:

“There’s a lot of people that don’t really care what happens to the horse. I mean, I care that my horse doesn’t (pause) die afterwards, you know? Because of that, I’ll ride competitive, but to a point, I’ll umm, never push him too much over his limit where like, some of these people umm (pause) they, they literally ride the horse till he drops. They’ve got too much money, too much [many] horses, I mean, how do you compete against someone like that who doesn’t put the horse first? I’ll never be able to compete against them because I’m not willing to do that. I actually want to take my horse home afterwards.”

From Hannah’s perspective, endurance riders who view the horse as a disposable commodity possess a distinct competitive advantage. Hannah had refused large offers of money for her horse Chance and had withdrawn from a recent race after sensing something was not right with him, to the chagrin of her fellow competitors. She had also decided to allow Chance to be administered intravenous fluids at a competition even though this precluded them from competing at a major, upcoming competition, all decisions which had limited their competitiveness. As the interview continued, it became increasingly evident that the bond between horse and rider had left Hannah questioning how to continue as a competitive rider when her chances of succeeding were reduced because of a strong horse-rider relationship. With evident discomfort, Hannah described the relationships she observed between other endurance competitors and their horses:

“I mean take the [group of riders]… they’ve got a partnership because they need the horse, but there’s no relationship, they don’t know the horses, they don’t train the horses, they only see the horse the first time at the competition, ride the horse till he drops, ride till he finishes, get off, go on. I mean that’s not a relationship.”

Although Hannah held significant concerns about her sport, she was a deeply competitive, motivated rider, still attempting to navigate a sport in which emotional detachment appeared to pay greater competitive dividends than forming strong attachments. Participants from other sporting disciplines also endorsed the possibility of competing successfully in the absence of a personal horse-rider relationship, though typically having a personal relationship was not seen as a disadvantage even if it was not necessarily always an advantage. Early in her career, one of Natalie’s horses had broken down after running hard across multiple cross-country courses, an experience that changed how Natalie behaved toward subsequent horses in her career. Yet like Hannah, she had experienced a backlash as a result of her ethos toward horses and horse-rider interaction:

“I wouldn’t go fast if I didn’t feel the ground was right or I’d just do the dressage and showjumping and not run the cross-country and I used [to], frustrate the hell outta [name] (laughs) because in those days you had to go fast at every bunfight otherwise you just weren’t a bloke, you know, you weren’t good enough, you couldn’t make a team because you just weren’t tough enough.”

Natalie’s decisions, it seemed, were interpreted by her teammates as compromising the performance-oriented masculinity of the sport, as well as jeopardising the success of any sporting teams of which she was a member. In fact, as she noted, her chances of making sporting teams had been minimised because of her decision to prioritise horse welfare over performance outcomes. Melissa also described suppressing her emotions to preserve a performance- rather than emotion-focused sporting identity that adhered to gendered expectations of elite athletes:

“…you tend to in those team situations, you don’t cry, you don’t, you can’t. You can’t be a female, you’ve gotta be a boy, “tough” and umm just play that kinda game.”

It may be that gendered notions of equestrian sport impact the judgments and behaviour of elite riders, particularly in instances where female riders are expected to prove their care and affection for horses will not make them any less competitive. Other participants had also felt pressure to prioritise performance success above all else, though their understandings of this did not necessarily refer to gendered constructions of sport. The autobiographies of Fox-Pitt [27], King [38], and Funnell [39] speak to the pressure riders experience in juggling the horse-rider relationship, horse welfare, the expectations of owners, team managers, and team-mates, and the desire to be competitive and make a living from equestrian sport, raising important questions around the management of horses in elite-level sport. 

Accounts of the relationship between horse and rider impinging negatively upon sporting success tended to contain an important link to the ethical treatment of horses in sport. The desire to protect and care for horses, as well as preserve the horse-rider relationship, made some riders less competitive in their sporting domain. The links between horse welfare, competitiveness, and the horse-rider relationship are important to consider given the centrality of horse welfare to the future of equestrian sport. In some contexts, tension emerged from an uneven playing field, with a disjuncture evident between the explicit and implicit requirements of elite sport and the value participants placed on their horses. In addition to the experiences of Hannah and Natalie, other participants described being criticized for refusing to do things such as showjump a tired horse, even when, in Georgina’s case, her decision not to compete did not affect her country’s rankings or chances of winning a medal. Empathy towards horses was always the right decision as far as Natalie was concerned, but she was aware of the potential drawbacks from a competitive point-of-view:

“The only downside to a partnership which to me isn’t a downside but maybe to somebody…that has a win-at-all-costs mentality, perhaps you know if you ride a horse like it is, like it is a winning machine, like it is just a, you know, a chattel, then maybe you can have the disregard that you need to beat a horse over a fence when it’s not right...that’s not me, you know, so maybe...if push comes to shove, in a situation like, I’m tough as anything and I’m tough as anything on myself, but there’s no way that I would make an animal do something that it would, that it would critically hurt it to do.”

Diana, too, expressed ambivalence about the influence of a strong partnership on performance in dressage, a perspective that reflects the concerns of Heuschmann [17] who writes that a revolution is needed in order to reintegrate horsemanship and by default, a sentient value for the horse into equestrian sport, noting that “too often, the [dressage] rider with the good, supple seat and the rhythmical and relaxed horse isn’t in the ribbons”. It is notable that the participants who had been criticised for prioritising horse welfare were female. Some female participants described the need to defend that their concerns for horse welfare did not make them any less competitive, even though this was clearly the case in some instances, while in contrast, Hannah was candid about the fact that her stance had made her less competitive. 

Importantly, sporting discipline may influence the impact of the partnership on performance outcomes. For example, endurance rider Lara reflected on the moment-by-moment synchronicity required of dressage dyads, noting, “They have to be so attuned to each other throughout the whole competition…they need to be with each other at every single step”. This aligns with Coffin [93], who speaks of dressage as the outcome of horse and rider “moving as a unit, in one balance”. Lara was ambivalent about the form of partnership required in her own sport of endurance, however, describing how she commonly competed horses with whom she had not worked before, suggesting that the partnership may be less consequential for endurance dyads, yet several dressage riders argued that harmony between horse and rider is not always recognized or rewarded in their sport. A former elite level, internationally successful eventer and current amateur showjumper, Lucy saw a partnership as critical for three-day eventing but less central in showjumping:

“For me as an eventer, I have to have, I feel like I need to have probably, compared with my current showjumping experience…a deeper more meaningful, longer term relationship because there’s just a lot more stuff going down and I don’t know that that’s the same for all eventing riders you know, cos a lot of them can hop on and go, but for me personally, I need to have a really good relationship with my horse and I can communicate at various different levels with it.”

In a similar vein, eventing rider, Pippa Funnell [39] compares the ‘business’ of being an eventing rider with that of being an elite showjumper, as is her husband, William Funnell:

“William’s business operates differently from mine. I am reliant on owners keeping horses with me and, as I am completely driven to get to the top, my interest is in building up the partnerships. Buying and selling has never interested me. William would love to be the same, as he has the same ambitions, but showjumping is a completely different world from eventing. There’s so much more prize money and therefore the value of the horses is greater, so he can’t justify keeping many of the promising young horses he buys to produce. There’s a permanent cycle of change in his yard—which is why he-and many other British show jumpers—has not always enjoyed the success at top level that he deserves. In contrast, I had a collection of promising youngsters that I envisaged keeping for their whole careers” (p. 87).

Eventing rider William Fox-Pitt [27] also describes how it can take around six years to produce “a top competition horse” (p. 234), suggesting horse and rider may work together for a considerable period of time in eventing before reaching elite levels, while Kelly made a similar comment with respect to Grand Prix dressage dyads. This supports the findings of Wipper [48] in which a partnership between horse and rider was seen as fundamental to one’s engagement in eventing, while Coulter [21] suggests that the culture of elite showjumping aligns best with “a particular way of being which is more compatible with the ways in which men are socialised to think and act” (p. 180). The discipline-specific importance of horse-rider partnerships is also addressed by Gilbert and Gillett [94] in their research on polo players, in which participants indicated how the relationship between horse and rider may be more or less important depending on the demands of the sport concerned. Moreover, Coulter [21] found that although showjumping is dominated by women, at the highest levels of the sport men represent the majority of competitors. In addition, Coulter [21] suggests that professional riding requires different qualities from amateur riding and that these qualities appear to be gendered insomuch as men far exceed women in professional ranks of equestrian showjumpers. 

Where the judging of performances is subjective [95] such as in dressage, whether a connection is drawn between the partnership and sporting performance appeared to depend on beliefs about the accuracy of the judging system, an issue to which Diana alluded. Where a strong partnership was seen as antithetical to the achievement of sporting success, the way in which dressage is judged was also typically seen as flawed. This also mattered for dressage performances in three-day eventing, with Matthew noting wryly, “You’d like to see the judge appreciate what’s happening”. An emphasis on physical appearance in dressage may also create a disjuncture between visual appearance and the actual quality of a dressage performance, something Melissa eluded to, noting how unlike dressage, showjumping was “relatively black and white”:

“Dressage can definitely be covered up, manipulated, pretended to be doing umm umm you know, like Monty, like I pretended my way to tests to Four Star and got okay scores but it was just on the edge of me being a very good showman in the ring and creating something that wasn’t really there.”

Dressage performances could be elicited in a range of ways, even if a partnership makes for “the best presentation” (Heidi, dressage rider) in the arena. In less subjective disciplines such as showjumping, a more straightforward dynamic between performance success and the horse-rider relationship was apparent. Of importance, harmony, or in some instances, the appearance of it, was not always viewed as ideal. As Matthew noted, strong performances in some disciplines may necessarily appear disharmonious:

“Going back to cross-country, sometimes it doesn’t look harmonious, sometimes if you’re getting the time, which you’ve got to go really fast to do, it looks a bit ugly, a bit wild and a bit rough and not very nice but sometimes that’s what you need to get the job done.”

Harmony was a fluid concept and its implications for the horse-rider partnership differed across sports. Finally, the partnership appeared particularly valuable to para-equestrian participants, who noted their inability to achieve performances via other means, while vaulting participants were more ambivalent, but emphasized the importance of trust between horse, vaulter, and lunger, highlighting the unique relational dimensions of vaulting sports. 

Critical to this discussion of sporting performances and horse-rider relationships is a sense of what is considered a successful performance. For a number of participants, performance success was defined in accordance with the achievement of conventional markers of success, such as a high placing or mark, and/or winning a competition, however, a number of participants also defined success or performance outcomes in subjective terms. Indeed, it was not uncommon for participants to define sporting success without reference to competition outcomes, and according to their own personal experience of a performance. The most significant performance of a participant’s career was in some instances representative of a personal, rather than a sporting triumph, with Hannah highlighting the importance of completing an endurance race at the same competition where her horse had almost died one year earlier. Other subjectively defined performances occurred when a dyad achieved their highest score to date or completed a competition without repeating a previously made mistake. Funnell [39] (p. 241) writes “…results are only relevant to the horse you’re riding,” while Melissa’s understanding of sporting success incorporated a similarly subjective view of success:

“Successful doesn’t mean (pause) winning, you know what I mean? Successful means, you know, going out and scoring a percent better again in that test and you know that horse did the test before and didn’t grind his teeth, it’s a huge win on that horse, he went in there and his transitions down from walk he didn’t pigroot because he’s too high behind, he didn’t do it this time and just little things like that, I go, “Yep that was better, that was better” and that for me is successful for that horse.”

Performing well for some participants meant performing confidently and/or well in adverse circumstances and in such instances, the partnership may be critical. Both Hayley and Patrick described their best vaulting performances as occurring when they were able to move expressively and confidently in the arena because of trust in the horse and lunger. A memorable performance could also result from seeing a development in the partnership between horse and rider, even if this was not accompanied by competitive success. Para-equestrian Kate described a frightening experience in the competition arena when Max began to panic, yet despite the stressful nature of this experience, Kate reflected on it positively as she had sensed Max trying to remain calm and communicate with her despite his distress:

“He was trying so hard not to do the full-on blind bolt but I could feel, like where he would normally switch off completely and just go, I could feel that sense of “Oh God Mum, oh shit I’m scared, what am I supposed to be doing, I’m really scared, I want out of here but what do you want me to do?” like…you could really feel him trying to work that out and that’s really special. I think to have got to that stage, even with how fearful he still gets, to at least get to that stage is a huge leap for that horse.”

To Kate, this performance was “horrible performance-wise,” yet it was one of the dyad’s biggest [rides] “relationship-wise,” with horse and rider connecting despite the horse’s fear. To Kate, the performance encapsulated “how much faith he [Max] was putting in me and that relationship”. This moment of connection did not lead to a positive competition result, but it was a meaningful achievement, nonetheless. A subjective understanding of sporting success meant that even an Olympic gold medal could be experienced as less than a perfect or satisfactory performance, evidenced in the hesitation in Kathy’s voice when she described her Olympic gold medal winning performance as “ideal…I suppose…” with reference to having had “the odd mistake” during the performance. Similarly, Melissa recalled the lack of emotion she experienced at having won an Olympic medal:

“At the time standing up there getting that silver medal I was going, “Where’s that feeling?” going, “It’s not here.” I’m going, “What’s going on?” You smile, you smile cos there’s lots (of) cameras, you smile and you’re going, “Why aren’t I feeling it, why aren’t I feeling the love?” you know and it was just, because I’d made mistakes and it ate at me, it really did. Not to the point where I was going to give up and I was crying but umm maybe if I had’ve cried about it, it would’ve been better.”

As Melissa’s account reflects, extrinsic success may not always be experienced as anticipated. Claire also recalled a successful competition result as unreflective of the true nature of the experience due to the pressure faced from team officials to change her horse’s performance style:

“Your hierarchies want to make everything better, but I got to the Olympics on that horse the way he was….but they wanted to make it all better and bigger and better and yeah, it was too much, shouldn’t have happened and then the horse started to go downhill from there and I started to lose confidence in them so you know, that was my worst, I mean I still got third or fourth and [country] still won the [competition name] so I mean, the placing actually doesn’t show the whole disappointment of it, but yeah.”

Furthering the nuances around how performance success may be regarded, extrinsic sporting achievements were not always accompanied by intrinsic satisfaction in one’s performances, nor did they necessarily represent something meaningful to the rider. Jim, for instance, described a particularly sensational performance that he did not “identify with, but it sure amuses other people”. In an equally nuanced fashion, Lara described one of her best performances as one in which she worked as a team with other riders from her country. They finished last in the competition but brought their horses home “safely”, worked well together, and enjoyed the ride. Moreover, during the ride Lara recognised “a mental switch” in her horse that made her realise the competitive heights still ahead of them but within their reach, an exciting personal outcome, while Abbey’s best performance was outlined without reference to the actual outcome, but in context to how the performance was experienced, “completely concentrated and ahh completely umm harmonious”. A rider may experience synergy with a horse, where the horse jumps clear and is “mentally really good” but such performances do not necessarily always translate into competitive success, thus whether the performance is experienced as successful or not in such circumstances depends in part on the rider’s goals and values. In reflection of this, Georgina was emphatic about her satisfaction with an Olympic performance that did not meet expectations:

“Everybody said, “Gosh I’m so sorry,” and I said, “I’m not, he was absolutely amazing, he did the best possible three phases he could do at the one time in our lives when we really needed to do that and five other people did it better.”

Georgina’s matter-of-fact account reflects the resilience participants regarded as central to enduring success, while also reflecting her emphasis on judging performances according to the strength of the partnership. It was not her Olympic medal winning performance that Lucy described as the most moving or ideal performance of her career, but instead another competition at which she experienced the “most perfect, perfect ride cross-country”:

“It was just like, so in sync, it was so perfect, just everything happened right and I came in right on time on the clock and it was just...perfect. Umm smooth and rhythm, and just communication and (pause) such a buzz.”

What occurs during positive sporting performances may be difficult to describe. Jim referred to high-quality performances as representative of “a beautiful, beautiful synergy between psychology and physiology”. In some circumstances a powerfully experienced physical and psychological performance collided with the achievement of extrinsic success, culminating in an experience that was both personally and professionally rewarding, as Elisa described in relation to her gold medal winning dressage performance:

“It was like, in a flow, everything just….ahh the both, the Special and the Kur, those both competitions in [city name] with him, that was like, you know, it was easy and it didn’t take any effort and we were one team. I didn’t even know there were people around, it was just him and me…you know, the feeling, it’s ahh once in a while you get a feeling and you think, “Ahh that was my best test,” but with him ahh you know it ahh all fell together, it was the two best tests and ahh Olympic medal and the whole atmosphere with it.”

Succeeding under pressure and experiencing an emotionally engaging synchrony between oneself and one’s horse may result in a sublime performance that reflects a flow state Csikszentmihalyi [96]. Mechanically correct and superlative performances may be differentiated from each other, with a partnership typically accompanying the latter, but not necessarily the former, as Amy describes here:

“If you don’t have a relationship...I mean you can go in and do the movements but… I don’t really know that it can meld into something that’s beautiful where you’re both flowing together. …somebody can go out there and ride a technically correct performance and that’s not the same.”

These findings suggest that competitive success may be measured via extrinsic or intrinsic factors and in either static or dynamic terms, with constructions of success influencing how, or if, participants connected the partnership with performance outcomes. It must be noted, however, that although participants often described sporting performances subjectively, when asked to consider the link between the horse-rider relationship and sporting performance, it was evident that ‘success’ was typically conceived in conventional terms, particularly when comparing the competitive orientation of different equestrian athletes. Understanding what is considered successful is clearly relevant to forming an understanding of how and why the horse-rider relationship affects sporting outcomes. Definitions of success provide an insight into the sporting orientation of equestrian athletes, while these definitions may impact how horses in elite sport are managed.

### 3.2. Investing Emotionally: Managing (De)attachment to Horses

Forming emotional attachments to competition horses engendered strong sporting performances in a multiplicity of ways, and in many cases, participants reported experiencing deep emotional bonds with their competition horses, particularly after working together for a significant period of time. Emotional attachment was not uniformly positive with respect to competitiveness at the elite level; participants described emotional detachment as a protective mechanism that could serve to mitigate the stress and risks of elite sporting participation. Following his first advanced competition win since retiring, Mark Todd [33] (p. 186) described having to wait while the course was “held” due to the death of another rider on-course:

“Eventually, the competition was restarted…and I won my section…That must sound extraordinarily hard nosed to people outside the sport, but it doesn’t mean that none of us care what happened. It’s just that riders develop a way of dissociating themselves from such an incident, or else none of us would be able to carry on.”

Distancing from the risks of equestrian sport with respect to the horse’s mortality was also sometimes necessary, yet even highly successful elite athletes struggled with this. Pippa Funnell [39] writes of being “shattered” (p. 64) at the loss of one of her young horses, describing how, “like all my horses, I’d got attached to him” (p. 64), while Todd [33] (p. 81) describes the “anguish” he felt when his horse, Face the Music, broke his leg during a cross-country round. Participants needed to be competitive with multiple horses, and this meant that their emotional energies were distributed accordingly. They also second-guessed their own emotional investments in horses because of concerns around the instability of third-party ownership arrangements, as was the case for Ethan. A number of participants were reliant on owners to purchase horses of the caliber they required to be competitive and in some cases this had emotional consequences for the intimacy of the horse-rider relationship. Para-dressage rider, Ellen made this clear when describing her current mount:

“The horse that I’m riding here umm I don’t have him, I have to drive five hours to ride him, so I ride him several times a month. He belongs to a friend of mine who competes in the Grand Prix, she’s planning to do the Pan Am Games selection trials with him next year, and she’s very graciously lent him to me but in no way is he mine.”

While failing to form a healthy relationship could be an impediment to performance success and enjoyment of equestrian sport, in some instances riders were advantaged by not having formed a close bond. A bond created emotional risks, and risks could readily morph into a threat to the relationship if the rider-owner agreement broke down over time. On the verge of losing the ride on two elite horses within a short space of time, Pippa Funnell [39] writes: “This was the first time I had to face up to selling a horse when I desperately didn’t want to. Of course, others had been sold before, but those sales had been the results of pragmatic joint decisions by myself and the owner that they weren’t going to do the job” (p. 141). As noted earlier, following the success of gold medal winning dressage horse, Valegro, every performance success increased the risk of Dujardin losing the partnership with him, creating an invidious performance environment for the dyad [97]. Although Funnell and other elite riders (cf. [27,33,70]) were cognizant of the financial concerns of elite horse owners and were appreciative of the support they received from owners, losing a competition horse could be both personally and professionally devastating. Such losses also served as a reminder of the limits to a rider’s agency and control over their own livelihood. With repeated losses, a certain measure of emotional self-preservation appeared to emerge for participants as a coping strategy in the face of great uncertainty.

Deliberately not naming or nick-naming a horse served as an emotion-management strategy. Where a transient horse-rider relationship was anticipated, emotional distancing was attempted through avoiding the intimacy that accompanies the attribution of a name, as Jane describes:

“If you have a horse where you think “Yeah it’s a turn-over horse” quite often I don’t...I mean I don’t know, all my horses have names and nicknames, I’ll have a horse that I’ll have him training or to sell and I’ll just call it the brown horse, so you don’t have that investment in the emotional side of it, and you ride those horses quite clinically I guess, it’s all about producing a product quickly.”

Naming horses may play a critical role in determining the emotional limits of the horse-rider relationship, with boundaries around certain horse-rider relationships facilitative of a competitive, business-minded approach to sport. The act of naming, Strauss [98] writes, directly affects the extent to which a sense of knowing exists between individuals, and in naming an animal or object, a sense of how to act and what to expect from that which has been named is indicated. Furthermore, as Sanders [77] notes, implicit in the act of naming is the attribution of individuality and identity, while non-naming practices tend to facilitate emotional distance, hence their common use in animal shelters where animals may be euthanized [99]. In the current study, individualising horses through naming practices appeared to foster an emotionally connected relationship and helped participants to construct a biographical understanding of individual horses.

Connecting emotionally with performance horses carries certain risks akin to the emotional experiences of scientists working with laboratory animals in Phillips’ [100] research, in which the practice of non-naming allowed participants to consider laboratory animals as a form of scientific equipment, separate and distinct from pet animals. Phillips [100] described the decision not to name laboratory animals as “an act of resistance to the social construction of individuality” (p. 129), enabling the laboratory scientist to construct a reality that minimizes the existential threat to the imagined self that might arise out of dissonance caused by engaging toward animals in ways that may cause them pain, harm, even death. Likewise, equestrian athletes must manage their feelings about endangering horses through sporting pursuits, particularly in sports such as endurance or eventing.

Non-naming practices were most commonly directed toward horses that riders knew they would only train or compete for a short period. In longer term relationships, maintaining an emotional distance was neither possible nor desirable, as many participants noted. Indeed, while there may be some competitive benefits to emotional detachment and participants sometimes felt they would be more competitive if they were able to maintain an emotional distance from horses, they also expressed a desire to remain emotionally connected to horse(s) and sporting activities. Naming, nick-naming, and personalising horses in intricate ways were common attributes of participants’ dialogues about horses. These findings suggest that managing emotions in an equestrian sporting context is a particularly complex activity, with both attachment and detachment potentially facilitative of sporting success. The conditions under which horse and rider interact and the level of control riders have over these interactions may have implications for equine management, horse-rider relationships, and the ethics of equestrianism.

## 4. Discussion

Largely speaking, the relationship between horse and rider was experienced as a vital part of performance success in the current study, in keeping with Pretty and Bridgeman [23] and Visser et al. [25]. However, ambiguous and contradictory views were expressed. Technical skills, physical strength, and contextual factors such as sporting discipline may in some instances outweigh the importance of a partnership, with some elite performances viewed as a product of talent and ability rather than a bond between partners. Some performances at the highest level of the sport were achieved in the absence of a partnership. Those participants who placed value on the relationship between horse and rider conceived it as a form of competitive currency; a way of compensating for other abilities, such as technical skills. In referring to deep attachment between horse and rider as a potential competitive disadvantage, participants brought into question the status of horses and the horse-rider relationship in modern equestrian sport. Urban myth, popular literature, and past scientific research have largely provided support for the notion that a strong, trusting relationship between horse and rider is integral to performance success [47,48], while the idea that those horse-rider dyads who possess the closest relationships are the most successful carries intuitive appeal. Yet, as Dashper [16] has suggested, the modern sporting environment places significant pressure on the horse-rider relationship, and a range of factors may affect the success of equestrian dyads.

While some felt strong performances could occur in the absence of a partnership, they nonetheless endorsed the partnership as instrumental in enhancing sporting performance. This supports Beauchamp and Whinton’s [101] research on self- and other-efficacy in equestrian sports, whereby confidence in one’s horse, a factor likely associated with a strong horse-rider relationship, was related to riders’ self-efficacy beliefs about themselves, with both self- and other-efficacy related to and predictive of enhanced performance. The subjectivity with which participants appraised their own performances also supports Evans and Franklin’s [102] argument that dyadic equestrian performances are experienced via the rhythm of horse-rider action, not simply as a manifestation of training. A number of participants described these harmonious exchanges as a primary motivation for their sporting participation, in keeping with Evans and Franklin [102], and with Kuhl’s [103] research on human-sled dog dyads.

Given the unpredictability of performances even for the most successful equestrians, investing in the horse-rider relationship and other aspects of the sport outside of competition outcomes may serve as a protective factor, moderating disappointment and distress should sporting failures occur. Participants described how a strong partnership could lead a dyad to exceed their individual ability levels, particularly where a horse lacked ability, but the horse-rider relationship was strong. Research on human sporting teams has suggested a similar pattern with chemistry or teamwork viewed as the catalyst for the success of less talented teams, while exceptionally talented sporting teams are on occasion unable to perform together as a team despite the ability of individual team members [104].

That the partnership was not the only factor, or even necessarily the most important factor influencing performance outcomes in elite equestrian sport, supports Beauchamp, Jackson, and Lavallee’s [105] contention that skill level, an athlete’s physical state, and genetic considerations greatly impact team-based sporting performances. Beauchamp and Whinton’s [101] study of relational efficacy in horse-rider dyads revealed that beliefs about one’s own abilities, as well as beliefs about the capabilities of one’s horse, explained unique variance in riding performance outcomes, with confidence in one’s horse predicting a unique amount of variance in performance scores when the effects of self-efficacy beliefs were controlled. Given that the quality of the relationship between horse and rider, as well as the knowledge a rider has of the horse, may affect a rider’s other-efficacy beliefs, it may be that the relationship between horse and rider plays an important role in mediating the impact of other-efficacy beliefs on sporting performance for equestrian athletes. In any case, Beauchamp and Whinton’s [101] work together with the findings of the current study suggest that interaction between horse and rider and the beliefs riders hold about their horses and their relationships with them may impact performance outcomes and how horses are managed.

Participants also described how in certain contexts, a strong partnership may be an impediment to performance, or in some circumstances, may not manifest in positive performance outcomes because other factors that influence performance are more influential, such as the perception of judges. Although it was seen as more likely to lead to short-term, rather than long-term success, participants challenged the notion that domination is an entirely unsuccessful way of relating to horses and achieving competitive success [106,107]. Where an athlete’s career is oriented exclusively around the achievement of performance success, domination-based relationships may be an effective and economical means of participating in equestrian sport. Where participants sought to form meaningful, trusting relationships in which a range of outcomes beyond performance success matter, partnering was viewed as the most economic and effective means of participating in equestrian sport [106,107], even if this meant performance goals took longer to achieve.

Distinctions between the partnership-performance connection were drawn in accordance with sporting discipline. Vaulting participants emphasized the role of technical skills alongside trust in both the horse and lunger, emphasizing the three-way dynamic between vaulter, lunger, and horse, while harmony was considered less relevant to eventing and showjumping than to dressage. An ambivalent image of the partnership-performance connection was drawn by endurance participants. This is in keeping with recent scrutiny of the sport with respect to doping, horse fatalities, and other sporting scandals [108] that some have argued create an uneven playing field and bring into question the moral ethics of endurance sports [109]. Notably, while this study sought to examine the connection between performance and the horse-rider relationship across a range of equestrian disciplines, no participants from western sporting disciplines such as reining, or participants from the sport of carriage driving, participated in the study. Future research should be conducted with these sporting cohorts to address this gap in the literature.

The most complex connection between the partnership and performance emerged in context to the sport of dressage. A primary source of ambiguity in relation to dressage was the judging of performance, horse-rider harmony in particular, in keeping with extant research [44]. These concerns may not be unwarranted, nor are they isolated. As Bryant [110] notes, controversy around dressage judging is a perpetual issue, and as McLean and McGreevy [19] note, while it may serve the public image of equestrian sport to try to ensure that successful competitors appear to also enjoy a harmonious relationship, what is required to actually succeed as an equestrian athlete may not always conform to this image.

Dressage participants described how harsh, forceful training practices were sometimes rewarded by judges, creating tension between the ethical and relational prerogatives of horse-rider interaction and the need to achieve competitive success. This ambivalence and concern may reflect the controversy over dressage riding practices such as unnatural gaits [110], hyperflexion [111] and forceful training methods that encourage unnatural movement, but are nonetheless often rewarded competitively [17,112,113]. Concerns over nationalism, wherein European riders tend to uniformly outscore other countries, and the halo effect, where top riders may receive a strong mark despite an average performance [110], something Georgina described experiencing because of her profile as a highly successful rider, may have also contributed to participants’ diffidence about the partnership-performance dynamic [110].

Participants were uncertain as to whether judges were accurate or actually even recognised quality performances, partially corroborating research by Stachurska et al. [44] that found inconsistencies and bias in dressage judging practices. In addition, research by Wolframm et al. [46] suggests that judges show certain visual fixation patterns that may explain observed inconsistencies in judging. Wolframm [45] has argued that the perceptual demands of judging dressage are significant, making it difficult to accurately and consistently examine dressage performances, though research by Bridgeman and Pretty [23] suggests judges are capable of accurately rating horse-rider harmony. Given that dressage is one of the few equestrian disciplines in which harmony between horse and rider is directly examined, these findings bring into question how readily identifiable horse-rider synchronicity is to an outside observer, as well as how dressage judges may go about measuring such a complex and subjective construct.

Performance success may be conceptualized in both subjective and objective terms, according to the findings of this study. Participants described the partnership-performance link in context to objective markers of success for other riders, yet often defined their own performance successes and failures subjectively. The subjectivity with which participants defined their successes reflects research findings by Durand-Bush and Salmela [114] on highly successful elite athletes in which the process as opposed to the outcome of sport was focused upon. It also supports the findings of a qualitative study on flow states in non-elite older athletes by Jackson et al. [115] that revealed performance success was subjectively interpreted and defined by participants. For equestrian athletes, achieving a harmonious unity with horses in which mutual focus and even synchrony of physical bodies [116] are experienced may result in a unique type of ‘peak performance.’ Participant accounts suggest that both extrinsic success and experiences of oneness and flow between horse and rider may be crucial to sporting motivation, as well as to participants’ continued participation in equestrian sport.

Although emotional attachment featured strongly in participant narratives, remaining emotionally distanced from horses in certain circumstances could facilitate a performance-based focus in equestrian sport. Where participants were aware that their interaction with a horse constituted a short-term business-oriented arrangement, remaining emotionally distanced was facilitated through (non)naming practices. This finding supports Myers’s [117] contention that naming inherently creates a sense of the individual, while non-naming may facilitate an instrumental approach to animals and this has ramifications for the management of horses in ‘business’ contexts, such as elite sport. Research by Irvine [118] suggests that the act of naming is central to affectionate, human-pet interaction and encourages the emergence of animal identity, while Dashper [16] points to the link between giving horses “stable name[s]” (p. 357) and attributing personal qualities to them, highlighting how working with horses that are owned by another party can create ambivalence in horse-rider relationships.

Emotional detachment in some circumstances was interpreted by participants as a competitive advantage, in other circumstances it constituted a competitive disadvantage, with inconsistency of contact between horse and rider and the presence of other figures in the relationship inhibiting the formation of a bond between horse and rider. The relationship between emotional (de)attachment and sporting success is clearly complex and may depend on whether a strong attachment is conceived as necessary for success, which may not uniformly be the case in equestrianism. Many participants appeared to, as Arluke [99] (p. 145) contends, simply “get on with the business of the institution” but this often required a certain measure of emotional detachment that may have adverse consequences for equestrian dyads in other ways. Closer consideration must be given to the emotion-management strategies that equestrians adapt as a result of the physical and emotional risks implicit in elite equestrianism.

Gender dynamics may be pivotal to how success, and successful relationships, are positioned within equestrian sporting discourses. Although the sample of the current study predominantly comprised female riders, interestingly, a range of positions were expressed by these participants in context to emotional engagement. Some embraced the concept of building relationships with horses as central not just to success, but also to their identities as riders, and perhaps also to their identities as female-identifying athletes moving within a male-dominant sporting sphere. Other female participants described needing to distance themselves from the sense that they might be ‘too soft’ to ‘do the job’ of being an elite rider, especially in team-based competition. Here, upholding feminine norms was abandoned in the face of needing to reinforce the notion of oneself as a serious athlete, willing and able to embody a performance-oriented, traditionally masculine approach to participating in sport. Interestingly, male participants did not seek to consciously articulate their position as an athlete in relation to gender and did not uniformly appear to grapple with the same tensions around performance and gender as their female counterparts, though male participants were concerned by power dynamics between horse and rider and were conscious of the expectation of stoicism, a somewhat antithetical position to authentic, emotional engagement with animals.

One lens through which the stance of equestrian athletes towards sporting performance and animal welfare may be conceptualized is an ethic of care, namely, feminist animal care theory. This theory, originally articulated by Gilligan [119], espouses an ethical stance towards human-animal interaction wherein relationships and autonomy are considered central to the development of a psychologically contextualised version of morality in which care work and animal voices and standpoints are paramount [120]. Emotionality, care, and a relational focus reflect a set of values that have traditionally been associated with femininity, but not, as Gilligan [119] points out, in a manner that should be treated as reductive or lesser when compared to so-called rational theories of morality.

Insofar as feminism may be understood as an attempt to liberate all beings, animal and human alike, the articulation of a feminist care ethic may be instrumental in liberating female-identifying equestrian athletes from the sense that the care work they perform in relation to their horses and sporting engagement is a liability to their professionalism, or in any way secondary to the more utilitarian, business-oriented dimensions of being an athlete. A feminist care ethic may also be central to the liberation of male riders from the notion that being ethical towards horses requires them to inhabit an emotion-free subject position that meets patriarchal expectations and reflects traditional masculine norms, within and beyond the sporting context. Finally, and perhaps most important of all, Gilligan’s [119] feminist care ethic invokes a new model of relating to animals wherein female, human understandings of the experience of being “ignored, trivialized, rendered unimportant” [120] (p. 306), and the psychological implications of such experiences, are translated into an ethical approach to animals that prioritises care and challenges hierarchical, speciest attitudes towards animals.

Future research should investigate how or under what conditions sporting performance may be considered antithetical to ethical horse-rider interaction and/or emotionally distanced interaction. The role of non-riding duties also deserves further consideration insofar as time spent physically maintaining a horse’s wellbeing and welfare, including fitness training, grooming, and animal rehabilitation and care, appears pivotal in allowing riders to build a multifaceted knowledge of the horse. This knowledge may serve a protective function in a competitive environment, where, in essence, the more closely the rider is able to detect changes, emotional or physical, in the horse, the better the welfare and relational outcomes for the dyad. Yet for practical reasons, many elite riders employ grooms and stable staff to assist in horse management, and for para-equestrian riders, there may be physical limitations around the amount of groundwork they can do with their horses. In addition to this, other riders may have limited access to their mounts for geographic or economic reasons, particularly in the European context. Considering how best to manage these practical concerns while maximizing contact between horse and rider at the elite level may be critical to rider and equine safety, as well as performance and relationship outcomes.

The challenge of modern equestrian sport is such that while it may be ideal to be involved in all aspects of horse care and management and to spend leisure time with horses, for many elite riders this is not commensurate with running a business that revolves around horses. Yet as the ethics of equestrian sport continue to be scrutinised and calls are made for a paradigm shift to eradicate unacceptable and unethical practices in the sport [61], it seems appropriate that all avenues of bringing about this paradigm shift be considered. While only a marginal degree of support currently exists for the complete abolition of equestrian sports [18], global attitudes toward horses in sport are shifting, with questions increasingly being asked by animal rights advocates about not just horse racing, but all equestrian sports [121]. The following question from Arluke [99] (p. 145) carries uncomfortable significance for equestrian sports, “…what is it about modern society that makes it possible to shower animals with affection as sentient creatures while simultaneously maltreating or killing them as utilitarian objects?”

While the regulatory bodies of the equestrian industry appear to strive to ensure inhumane treatment of horses in sport is identified and punished [110], recent incidents involving the disciplines of endurance and eventing have sparked considerable controversy with respect to the actual impetus of the FEI in attempting to prevent these occurrences (cf. [109,122,123]). Course reform and other changes to equestrian sport have created a more welfare-oriented sporting landscape, but to a large extent the welfare of horses rests in the hands of their riders, a point conveyed by multiple medal-winning three-day event rider, Michael Plumb (as cited by Bryant [110], p. 140), who, with respect to the dangers of the cross-country phase in eventing, stated “…right now they’re talking about fixing the jumps, but I think the answer is basic horsemanship.” It is notable that changes designed to make equestrian sports safer, such as the short-format cross-country course and the omission of the steeplechase and roads-and-tracks phase in eventing, have not, as Bryant [110] points out, been overly successful in making equestrian sports safer for horse or rider.

## 5. Conclusions

This paper has addressed the connection participants drew between sporting performance and the horse-rider relationship. While a link was drawn between sporting success and strong horse-rider partnerships, ambivalence also marked participants’ conceptions, with some suggesting a partnership is essential to success, and others noting how a range of factors determine sporting outcomes. This suggests that the importance of the horse-rider relationship to performance success must be contextualized to determine how optimum management of horses aligns with sporting practices across equestrian disciplines. The partnership was also understood as detrimental to success in some circumstances. Equestrian athletes who take a detached, win-at-all-costs approach to equestrian sports were viewed as having a competitive advantage in some sports, although they were also considered less likely to maintain a successful long-term riding career. The connection between performances and the partnership differed according to sporting discipline, while participants defined success in subjective as well as objective terms.

The connection between sporting performance and strong horse-rider relationships was approached with ambivalence by participants. A strong partnership allowed participants to predict their horse’s behaviour and created confidence and a sense of connection and synergy that may be critical to sporting success, particularly given the highly-strung, sensitive temperaments of many elite horses and the anxiety transference that occurs in horse-human interaction [124]. Research has indicated that successful equestrians tend to experience less anxiety, and higher levels of self-confidence, positive thinking, determination and concentration than their less successful counterparts [125] and the horse-rider relationship appeared critical in moderating participants’ experiences of some of these factors. However, while a strong partnership could enhance performance, it was not always seen as essential for performance success, and in some contexts it was viewed as inhibiting success.

Sport has the capacity to be “both elevating and deflating, appealing and appalling, inspiring and disillusioning” [126] (p. 225) and in equestrian sports this may be particularly the case because of moral concerns around using horses in sport. Equestrianism only indirectly contributes to the development of society and revolves predominantly around human fulfilment and leisure, with the use of horses in sport not necessary in the way that using horses for transport was once necessary. As Midgley [127] contends, “ambivalence has always been central to most of the ways in which humans use animals” (p. 193). This was evident in the emotional and cognitive dissonance participants displayed when rationalising their own engagement in sporting practices that have the potential to lead to stress, harm, and even death for their equine counterparts. Those involved in equestrian sports commonly espouse their love of horses and concern for horse welfare, but this has the potential to create dissonance around engagement in equestrian sport [110], and this appeared to be the case for some participants in the current study.

Achieving performance outcomes is by definition of importance in an elite sporting context and as Eitzen [126] suggests, nowhere is the pressure to succeed greater than in an elite, professional sporting context. With this pressure comes an increased risk of unethical behaviour. That participants who experienced close horse-rider relationships and valued their horses highly sometimes felt less competitive as a result suggests that an examination of the formal markers of success in elite equestrian sports is required. Where the achievement of success sits in opposition to horse welfare and horse-rider partnerships, equestrian athletes are likely to face difficulties juggling horse welfare, competitiveness, the horse-rider relationship, and the expectations of team officials, owners, sponsors, and their teammates. Attention must also be paid to how athletes navigate a life-long career with horses that, for a number of participants in the current study, had already involved the death or serious injury of a horse due to sporting activities. Alongside these concerns lies the emphasis participants placed on partnering horses in egalitarian ways, providing support for the contention that equestrian sport has the potential to be “not only morally defensible, but desirable” [3] (p. 17), though this may require active attention to the moral rights of animals and the development of a sporting culture in which being competitive and valuing animals are congruent.

Given that re-integrating horsemanship practices into modern equestrian sport may be pivotal in strengthening horse-rider relationships, improving horse welfare and enhancing the competitiveness of equestrian dyads, it is worth considering different ways of incorporating less pressured forms of social contact with horses into the lives of elite riders, while the attitudes of modern riders also deserve closer consideration. Equestrian athletes must be encouraged to act as gatekeepers of equine welfare and well-being, and to encourage this, the consequences of acting in welfare-conscious ways towards horses must be aligned with performance outcomes. Several participants described how their engagement with horses and sport had improved since making the decision to seek income away from equestrian activities as this had allowed them to draw a clear distinction between work and working with horses. Access to alternate sources of income may help to preserve the personal fulfilment participants often associated with their relationships with horses and may also be facilitative of stronger, more engaged experiences of sport and sporting performances. Equestrian athletes should bear responsibility for the welfare of their horses. However, it is incumbent upon the broader sporting community to create a sporting culture in which animal welfare and equine management are inseparable from high-quality performance outcomes and ethical horse-rider interaction.

## Figures and Tables

**Table 1 animals-11-01352-t001:** Demographic characteristics shown as frequencies.

Participant Demographics	*N*	(%)
Gender	36	(100.0)
Male	5	(86.1)
Female	31	(13.9)
Primary riding discipline		(100.0)
Dressage	9	(25.0)
Eventing	12	(33.3)
Showjumping	1	(2.8)
Endurance	3	(8.3)
Vaulting	4	(11.1)
Para-equestrian	7	(19.4)
Country of Origin		(100.0)
Australia	16	(44.4)
United Kingdom	2	(5.6)
United States of America	6	(16.7)
Canada	5	(13.9)
The Netherlands	2	(5.6)
Germany	1	(2.8)
South Africa	2	(5.6)
New Zealand	2	(5.6)

## Data Availability

Data presented in the current study is not publicly available, in the interests of ensuring participant confidentiality given the potentially identifiable participant sample.

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
