# Peer review of "Symbiosis or Sporting Tool? Competition and the Horse-Rider Relationship in Elite Equestrian Sports"

_animals, 2021, doi:10.3390/ani11051352_

Round 1

Reviewer 1 Report

Dear Authors,

The work is very important however crucial information about exercise physiology is missing. 

The study seems very interesting and extremely important. However, some corrections are needed. The major weakness of the paper is that it is in some parts word choices seemed odd, making it read more like creative writing or a magazine article rather than scientific style. Also very often the Authors used “…” which is not so common used in scientific writing.

However, I am really glad that Authors wrote this article because the topic is really important.

Performance and Relationships in Equestrian Sports

In this paragraph I recommend to add some information about performance monitoring techniques. The most sophisticated examples are used in race training. Lactate blood concentration is the most valuable parameter which leads to calculate VL4 which is changing during training progress. In addition Authors should add the information about the most developed techniques which are nowadays used especially in race training such as infrared thermography which also correlates with lactate concentration. Also other parameters may be used in training monitoring such as. changes in PBMCs proliferation and activity. Thus, this information is missing.

Equine Welfare in Elite Sport

Paragraph about stress reaction should be added while it strongly influence on welfare of the horse athletes. Stress response after training sessions and competition is different and it is mirrored by changes in blood cortisol concentration. Thus information about this should be mentioned because it is important for horse welfare to show the full picture of the welfare issues in elite sport.  

Reviewer 2 Report

I applaud the author for initiating work in a very complex area, which impacts not only psychological health of the equestrian but certainly the well being of equine partners.

There are a few weaknesses of the study that the author should address.

  1. There is a very real possibility that the interviews discussed their horse-human with a social desirability bias - trying to please the interviewer with what they felt was a more acceptable horse - human relationship.  Conceivably the interviewees were also experiencing cognitive dissonance with their own attitudes and beliefs.
  2. The sample pool was heavily skewed to include female riders, despite the heavy prevalence of male riders at the elite level.  The interviewees even discuss gender bias in riding sports and need to act more in a masculine manner.  I would suggest the author refer to the feminist care theory in their discussion and certainly interview more male athletes in the future.  see line 930
  3. The author mentioned reining as a potential discipline pool to draw interviewee subjects but no members of this sport were included.  Riders in western disciplines may differ dramatically in their attitudes towards equine well-being and partnership (personal experience). Please refer to E.A. Lofgren, M.A. Tucker, B. Rice, M.A. Voigt, C.M. Brady,
    Does discipline matter? An analysis of equine welfare perceptions and beliefs in the context of horse show participation,
    Journal of Equine Veterinary Science,
    Volume 52,2017,Page 115,ISSN 0737-0806,
    https://doi.org/10.1016/j.jevs.2017.03.188.
    (https://www.sciencedirect.com/science/article/pii/S0737080617303064)
  4. The author may want to reference sports (in the US) that do use the horse solely as a prop or tool (ie collegiate equestrian competitions).

I suggest shortening the portion of the materials and methods that describe the authors background and complete elimination of the use of first person.  This can be tightened down without removing the context or lens through which the study was performed.

Specific comments: 

Line 17 - define instrumental approach

line 64 - describe what format changes to Olympic competitions has increased marginality of the sport

line 181 - I believe it should read "due to"

line 772 - should this be did not prevent?
